# EigenGuard: Backdoor Defense in Eigenspace

## Abstract

Deep Neural Networks (DNNs) have shown remarkable performance in various downstream tasks. However, these models are vulnerable to backdoor attacks that are conducted by poisoning data for model training and misleading poisoned models to output target labels on predefined triggers. Such vulnerabilities make training DNNs on third-party datasets risky and raise significant concerns and studies for safety. With an unauthorized dataset, it is difficult to train a model on such data without the backdoored behavior on poison samples. In this paper, we first point out that training neural networks by forcing the dimension of the feature space will induce trigger misclassification while preserving natural data performance. Based on these observations, we propose a novel module called EigenGuard, naturally trained with such a module will make neural networks neglect triggers during training on the unauthorized datasets. Experiments show that, compared with previous works, models with our EigenGuard can show better performance on both backdoor and natural examples compared with other defense algorithms.

## 1 Introduction

Deep Learning has achieved remarkable success in an array of tasks, including computer vision (Kirillov et al., 2023; Szegedy et al., 2016; 2017), speech recognition (Wang et al., 2017), and others (Brandes et al., 2022; Nussinov et al., 2022). With deep learning models finding extensive deployment in critical applications, their security problems, such as adversarial attacks, backdoor attacks, privacy concerns, and more, have gained increasing attention in recent years.

Among these threats, the backdoor attack is one of the most vulnerable threats which induces models to predict target classes when encountering triggers by poisoning the model's training dataset. To implement such an attack, adversaries inject backdoor triggers into machine learning models by introducing specific trigger patterns into a limited subset of the training data (Chen et al., 2017; Liu et al., 2018b). The primary aim is to forge robust connections between these covert backdoor triggers and designated target classes while keeping the model's original relationships between inherent features and their labels corresponding to natural images. Consequently, models that suffer from backdoor attacks demonstrate normal behavior when exposed to clean inputs but can be surreptitiously manipulated to predict the target class when confronted with inputs attached with triggers. Notably, DNNs have been identified as particularly susceptible to such backdoor attacks (Liu et al., 2018b). Furthermore, backdoor triggers, once integrated, tend to be elusive and challenging to detect or remove, presenting substantial security challenges to the realm of deep learning.

When attempting to train a clean model on unauthorized datasets, many existing methods try to fine-tune the neural networks on some additional datasets. However, such a method may not always be effective since the tuning datasets may not change any neurons corresponding to backdoor triggers when the distribution of fine-tuned datasets and backdoor datasets are dissimilar. Apart from that, how to access a clean subset for tuning is also an open problem. To deal with such problems, some works try to detect backdoor samples and purify the datasets Tran et al. (2018); Yuan et al. (2023); Gong et al. (2023). Then they try to unlearn or retrain neural networks to defend backdoors. However, the accuracy for detection is also not stable and may influence the final defense results. For example, the spectral signature method Tran et al. (2018) fails in most cases in our paper with the backdoor bench's setting. Furthermore, such methods need a second stage in their training methods, which is complicated and different from the widely used end-to-end training diagram. Therefore, how to develop an end-to-end defense method for backdoor attacks is worth exploring.

| | Clean Data | Unlearning | multi-stage Training | End-to-End Training |
|---|---|---|---|---|
| Finetuning | ✓ | × | ✓ | ✓ |
| Pruning | ✓ | ✓ | × | × |
| Splitting and Unlearning | × | ✓ | × | × |
| Splitting and Retraining | × | × | ✓ | × |
| EigenGuard (ours) | × | × | × | ✓ |

Table 1: Summary for the characteristics of current backdoor defense methods.

To tackle the above weaknesses, we try to design a defense method without additional datasets and training procedures called EigenGuard. It is a new module for deep networks and can defend against various backdoor attacks with a satisfying performance by natural training without the need for additional datasets. To clarify the difference between our method and other current methods, we summarized the characteristics of the existing state-of-the-art backdoor defense methods in Table 1.

In the following paper, we begin by revisiting the spectral characteristics of features during training. Initially, we observe that trigger features (backdoor features) tend to exhibit a concentrated behavior within the spectral space, particularly around the top singular values. Inspired by such findings, we propose our EigenGuard which can make the trigger feature ineffective with other natural features by forcing top $k$ spectral features to share the same scale during training. Then our EigenGuard can lift the scale of natural features when encountering poison samples and prevent the model from predicting trigger class only based on trigger features. As for natural examples, their performance is less susceptible to our EigenGuard because the effective feature for natural classes is rich enough to make correct predictions as our analysis shows. The experiments also demonstrate that our EigenGuard enjoys superior consistency and performance when compared to alternative defense methods, delivering enhanced results in many cases, especially on natural examples. In summary, our contributions can be briefly outlined as follows:

1. We find the useful features for backdoor images are centered at a low-rank space. Therefore, forcing a high-dimensional feature space will make backdoor images fail to attack.

2. We find that effective natural features are distributed in a high-dimensional space compared with backdoor features. Therefore, forcing various features will not influence the performance much.

3. We then propose a new module based on our findings. With the new module, neural networks can consistently defend against widely used backdoor attacks without additional data and other training techniques, which means users can train models with our EigenGuard safely with vanilla end-to-end training procedures.Furthermore, the natural accuracy of our method is also better than other defense methods.

## 2 RELATED WORK

### 2.1 BACKDOOR ATTACKS

The backdoor attack is a category of assaults that occur during the training of deep neural networks (DNNs). In this type of attack, attackers try to contaminate a portion of the training data by adding a predefined trigger and reassigning them as the desired target labels, which is known as the "dirty-label setting" (Gu et al., 2019; Chen et al., 2017). These trojan samples can either all be relabeled as a single target class (known as "all-to-one"), or samples from different source classes can be relabeled as distinct target classes (referred to as "all-to-all") (Nguyen & Tran, 2020). Subsequently, after the model's training phase, the attacker can manipulate models to predict the target labels by attaching triggers during testing.

Such attacks differ significantly from other evasion attacks, such as adversarial attacks (Biggio et al., 2013; Szegedy et al., 2014; Goodfellow et al., 2015). Backdoor attacks are focused on implanting a trigger into the model that is agnostic to both the input data and the model itself, posing a significant threat to the applications of deep learning (Goldblum et al., 2020). To avoid easy detection of incorrectly labeled samples, some attackers attach the trigger to samples from the target class, known as the "clean-label setting" (Shafahi et al., 2018; Turner et al., 2019; Barni et al., 2019).

In addition to simple forms like a single-pixel or a black-and-white checkerboard (Tran et al., 2018; Gu et al., 2019), trigger patterns can also take more intricate forms, such as a sinusoidal strip or a dynamic pattern (Barni et al., 2019; Nguyen & Tran, 2020). Recent attacks have made these triggers more natural (Liu et al., 2020) and imperceptible to humans (Zhong et al., 2020; Nguyen & Tran, 2021), rendering them stealthy and challenging to detect through visual inspection. Furthermore, powerful adversaries with access to the model can optimize the trigger pattern (Liu et al., 2018b) and even co-optimize the trigger pattern and the model together to enhance the potency of backdoor attacks (Pang et al., 2020).

## 2.2 BACKDOOR DEFENSE

**Defense with training data.** When users need to train a clean neural network on unauthorized datasets, the defenders usually try to detect and neutralize poisoned data. One widely used way to identify these poison data is designing some outlier detection methods with some robust statistical methods in either the input space or the feature space (Steinhardt et al., 2017; Koh et al., 2018; Diakonikolas et al., 2019; Gao et al., 2020). These robust statistics techniques facilitate the identification and removal of such anomalies, preserving the integrity of the training data.

Alternatively, researchers have explored various training strategies aimed at mitigating the impact of poisoned data on the trained model (Li et al., 2021b; Tao et al., 2021). These strategies include randomized smoothing (Rosenfeld et al., 2020; Weber et al., 2020), majority voting (Levine & Feizi, 2021), differential privacy (Ma et al., 2019), and input preprocessing techniques (Liu et al., 2017; Borgnia et al., 2021). By incorporating these methods into the training pipeline, defenders can enhance the model's resistance to poisoned data, ultimately reinforcing its security and dependability. However, such detection methods can not successfully detect the backdoor images especially when a large amount of images are poisoned. Therefore, the inaccurate detection will weaken the performance of the defense methods or model's natural accuracy.

**Defense with additional clean data.** When dealing with downloaded models with the potential of being poisoned, one possible way to purify the model is to fine-tune the model with additional clean images. Apart from such a simple method, one approach for enhancing the defense performance is to initially reconstruct an approximation of the backdoor trigger based on the clean subset. This can be achieved through adversarial perturbation techniques (Wang et al., 2019) or by utilizing generative adversarial networks (GANs) (Chen et al., 2019a; Qiao et al., 2019; Zhu et al., 2020). Once the trigger is successfully reconstructed, it becomes feasible to prune neurons that activate in response to the trigger or fine-tune the model to unlearn it, as demonstrated in previous work (Wang et al., 2019).

However, recent advances in attack methods have introduced more complex trigger patterns, such as dynamic triggers (Nguyen & Tran, 2020) or triggers based on natural phenomena (Liu et al., 2020), making reconstruction increasingly challenging. Some studies have explored trigger-agnostic repair approaches through model pruning (Liu et al., 2018a) or fine-tuning on clean data (Chen et al., 2019b; Li et al., 2021a). It's worth noting that these methods may suffer from significant accuracy degradation when only limited clean data are available, as observed in (Chen et al., 2019b).

## 3 REVISITING THE SPECTRAL BEHAVIOR OF NEURAL NETWORKS' FEATURE

### 3.1 PRELIMINARIES

The neural network, denoted as $h$, operates on input data $\mathbf{x}$. $f$ represents the head of the model $h$, the first residual stage for example if $h$ is ResNet, while $g$ refers to the latter part of the model. Thereby, $h$ can be formulated as the composition function of $g$ and $f$, that is $h = g \circ f$. Furthermore, we employ the notation $\mathbf{z} = f(\mathbf{x})$ to represent the intermediate features generated by $f$. Within these intermediate features, the singular values are denoted as $\sigma_i$, with the index indicating their scale sequence. Additionally, the label associated with the input $\mathbf{x}$ is represented as $y$ in subsequent discussions.

## 3.2 SPECTRAL ANALYSIS ON BACKDOOR IMAGES

According to former research (Feng et al., 2022), the dimension of the deep neural networks feature space will be much smaller than its design. Therefore, the models will leave some redundant dimensions during natural training. Since the feature of triggers will be much simpler than the image's natural structure according to former works, we would like to verify whether the dimension of features will influence the model's backdoor training behavior. Firstly, we try to assess the effectiveness of the poisoned sample by progressively eliminating the dimension of the feature space after the first residual stage in ResNet-18 by setting the top singular values of features SVD decomposition to 0 during the training process. In addition to reducing the dimension, we also try to increase the feature dimensions by lifting the original small singular values and generating new features for training and testing. The attack success rate for these two scenarios is drawn in Figure 1 (a) and (b).

The figure clearly illustrates a significant decline in the model's attack success rate when introducing additional subspace to the original feature space, whereas performance remains unaffected when removing subspace during training. One plausible explanation for this observation is the limited effective subspace associated with the trigger. This suggests that the trigger features are distributed in a low-dimensional subspace. Consequently, when we compel the neural network to acquire more features, the natural features contained in the samples will be extracted by the model and the natural features will lead the neural networks to predict their true class instead of the target class.

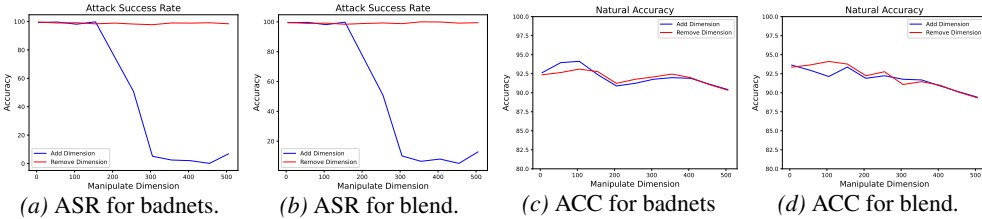

| *(a)* ASR for badnets. | *(b)* ASR for blend. | *(c)* ACC for badnets | *(d)* ACC for blend. |

Figure 1: Attack success rate (ASR) and natural accuracy (ACC) of different poison methods when adding dimensions or reducing dimensions.

## 3.3 SPECTRAL ANALYSIS ON CLEAN IMAGES

In this section, we extend our exploration to uncover the relationship between the dimensions of natural features and the model's natural performance from the spectral perspective. Just as we did with backdoor features, we evaluate the effectiveness of natural features by incrementally removing the subspace correlated with the top singular values (reducing dimensions) and lifting the small singular values to incorporate more subspace (adding dimensions) during the training process. The results are drawn in Figure 1(c) and (d), which illustrate the accuracy of clean images.

As depicted in the figure, it is apparent that the model's natural accuracy remains virtually unchanged when introducing additional subspace to the feature space, and there is no large drop in performance when removing subspace during training. One plausible explanation for this behavior is that the effective feature subspace corresponding to the true label is already sufficiently large. Consequently, the model consistently manages to extract valuable features from natural samples, regardless of the size of the feature subspace enforced during training.

To evaluate the above reason, we also calculate the effective rank (Roy & Vetterli, 2007) for features obtained after ResNet-18's first residual stage with respect to natural and backdoor inputs. The calculation of effective rank (ERank) for matrix $\mathbf{A}$ can be formulated as follows:

$$\text{ERank}(\mathbf{A}) = -\sum_i p_i \log(p_i), \tag{1}$$

where $p_i = \frac{\sigma_i}{\sum_i |\sigma_i|}$ and $\sigma_i$ denotes the $i$-th singular value of matrix $\mathbf{A}$. The results are listed in Table 2. One can see that the effective rank for natural examples is higher than the ranks for backdoor samples. Thereby, the dimensions of backdoor feature space are smaller than natural feature space.

Apart from the effective rank, we also draw t-SNE figures on natural examples and backdoor examples by first projecting their output features on its eigenspace with respect to the first ten singular

|  | Blend | BadNets | SIG | Clean Label |
|---|---|---|---|---|
| **Backdoor** | 2.7 | 3.5 | 4.0 | 3.1 |
| **Clean** | 3.6 | 4.0 | 4.4 | 3.5 |

Table 2: Effective rank of backdoor samples and clean samples for a naturally trained ResNet-18 on CIFAR-10.

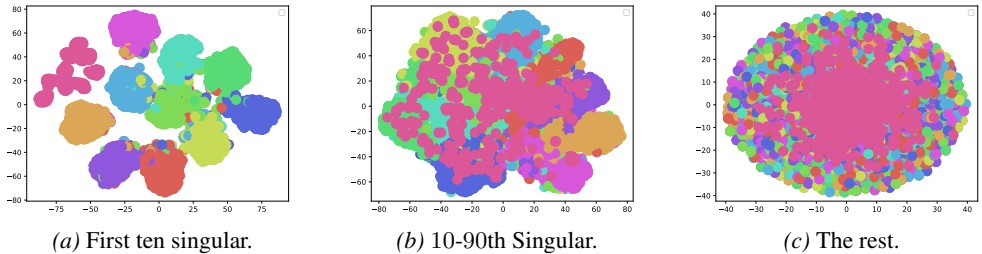

*(a) First ten singular.*  *(b) 10-90th Singular.*  *(c) The rest.*

Figure 2: t-SNE map on neural networks' output features with respect to different eigenspace for backdoor and clean images. The "pink" dots represent the backdoor samples while the other colors stand for different natural samples with different classes.

values, the next 90 singular values, and the rest of the singular values. We treat backdoor classes as the 11-th class on CIFAR-10, and results are shown in Figures 2.

From the figure, one can see that the t-SNE map for the features with respect to the top singular values is discriminative for both backdoor ("pink dots") and natural images ("other colors"). However, from the middle t-SNE figure, one can see that the pink dots represent backdoor images distributed uniformly in the space and overlap with other color dots. Thereby, the network cannot classify these samples as trigger classes since they are similar to samples belonging to different natural classes. However, one can see that the natural images can also be classified although some samples are overlapped. The third figure shows that samples cannot be classified with rest features. The above figures validate our findings that backdoor samples' dimensions for useful features are smaller than natural samples.

### 3.4 EIGENGUARD FOR BACKDOOR DEFENSE

Building upon the insights gained from the preceding analysis, it is clear that models can maintain their natural performance while effectively neutralizing the impact of backdoor connections when forcing the feature dimensions to be large. Leveraging this critical understanding, we can propose our defense mechanisms against backdoor attacks by seamlessly integrating an EigenGuard module within neural networks. Our EigenGuard approach is illustrated in Algorithm 1. As one can see, the algorithm can scale the features belonging to the top $k$ eigenspace. Therefore, the effective rank of the feature will increase when $k$ gets larger. Then the backdoor sample classification can be misled by natural features lifted by our EigenGuard module based on our former analysis. We also make a toy model for the theoretical analysis in the following section.

### 3.5 THEORETICAL UNDERSTANDING ON EIGENGUARD

In this section, we are going to theoretically analyze the effectiveness of our EigenGuard module with a toy model for binary classification. First, we define the latter model $g$ for classification as $\mathbf{w}^\top \mathbf{z}$ with $\mathbf{z} = f(\mathbf{x})$ and $\mathbf{w} \in \mathbb{R}^{5 \times 1}, \mathbf{z} \in \mathbb{R}^{5 \times 1}$ and $\mathbf{w}$ has been normalized. Furthermore, the output features $\mathbf{z}$ for each sample can be viewed as the composition of the following five eigenvectors $[\mathbf{f}_1, \mathbf{f}_2, \mathbf{f}_3, \mathbf{f}_4, \mathbf{f}_5]$ obtained by the projection on each eigenspace and each sample $\mathbf{z}_i$ can be formulated as $\mathbf{z}_i = \sum_i \alpha_i \mathbf{f}_i$ and $\mathbf{f}_5$ denotes the features vectors for triggers while the others are natural features. We also set $\sum_i \alpha_i = 1$ for convenience. Then we have the following results:

**Proposition 1.** *For features $\mathbf{z}$ corresponding to input $\mathbf{x}$, if the latter model $g$'s weight $\mathbf{w}$ can classify natural features ($\mathbf{f}_1$ to $\mathbf{f}_4$) to be positive (natural class) and backdoor features ($\mathbf{f}_5$) to be negative*

---

**Algorithm 1:** Inference procedure for neural networks with our EigenGuard.

---

**Require:** Input $\mathbf{x}$, head module $f$, remaining modules $g$, filter factor $k$, history feature $His$, Momentum $\mu$.

**Ensure:** Get the output feature $\mathbf{z}$ obtained from our EigenGuard with respec to $f(\mathbf{x})$.

Updated history features:

$$His = \mu His + (1 - \mu)f(\mathbf{x})$$

Concatenate current features with history features:

$$feat = [His, f(\mathbf{x})]$$

Do Singular Value decomposition on $feat$:

$$\mathbf{U}, \Sigma, \mathbf{V} = SVD(feat)$$

Filtering features with a spectral filter:

$$\Sigma[: k] = \sigma_k, \qquad \Sigma[k + 1 :] = \sigma_k * 0.001.$$

Get new features with the same size as $f(\mathbf{x})$:

$$\mathbf{z} = \mathbf{U}(\Sigma_{new})\mathbf{V}^\top[: f(\mathbf{x}).size(0), ..]$$

**return z**

---

*(target class) with output scales lies in $(\beta, 1)$ as the following states:*

$$\mathbf{w}^\top \mathbf{f}_i \in (\beta, 1) \text{ if i} = 1, 2, 3, 4, \qquad \mathbf{w}^\top \mathbf{f}_5 \in (-1, -\beta), \tag{2}$$

*Then the network will predict the input as the natural class instead of the target classes if the composition of the output feature obeys the following condition:*

$$\sum_{i=1}^{4} \alpha_i > \frac{1}{1 + \beta}. \tag{3}$$

From the above proposition, one can see that if the composition of natural features is strong enough, the neural networks can correctly predict the natural class instead of the target class. However, such a scenario may not easily happen since backdoor samples may be learned to let $\sum_{i=1}^{4} \alpha_i$ go small for vanilla models' features during training. Then the features' natural components will be weaker and the neural networks will make predictions as the target based on backdoor features. Fortunately, by adding our EigenGuard with $k = 5$ in the neural network, the components of natural features will not vanish with $\sum_{i=1}^{4} \alpha_i = 0.8$ even on backdoor samples. Moreover, we have the following:

**Remark 1.** *In the above model, the model can always predict the correct natural class instead of the target class with our EigenGuard and $k = 5$ only if $\beta > 0.25$.*

We need to note that the above condition can be easily achieved during training as $\mathbf{w}$ and $\mathbf{f}_i$ are normalized. Therefore, our EigenGuard can neglect backdoor triggers and make true predictions.

## 4 EXPERIMENTS

### 4.1 EXPERIMENT SETTINGS

**Backdoor attacks and settings.** We adopt six state-of-the-art backdoor attacks: 1) BadNets (Gu et al., 2019), 2) Blend backdoor attack (Blend) (Chen et al., 2017), 3) Clean-label backdoor (CLB) (Turner et al., 2019), 4) Sinusoidal signal backdoor attack (SIG) (Barni et al., 2019), 5) WaNet Nguyen & Tran (2021) and 6) SSBA Li et al. (2021c). To ensure equitable comparisons, we adhere to the default configurations outlined in their respective original papers, including trigger patterns, trigger sizes, and target labels. The evaluation of both attack and defense strategies takes place on the CIFAR-10, CIFAR-100 (Krizhevsky & Hinton, 2009) and GTSRB datasets with $10\%$ poison rate employing ResNet-18 (He et al., 2016) as the underlying model. During the training of the neural networks, $90\%$ of the training data is utilized, with the remaining $10\%$ being used, in whole or in part, for defense purposes. We set $k = 20$ and momentum to be $0.7$ for our defense and we also use 100 training samples $(< 1\%)$ with the lowest loss score to make the SVD decomposition more accurate. Additional implementation of attacks can be found in the Appendix.

**Backdoor defenses and settings.** We compare our proposed EigenGuard with 5 existing backdoor defense methods: 1) standard fine-tuning (FT), 2) Adversarial Neural Pruning (ANP) (Wu & Wang, 2021), 3) Anti-Backdoor Learning (ABL) (Li et al., 2021b), 4) Neural Attention Distillation (NAD)

| Data | Types | | None | FT | ANP | NAD | SS | ABL | EigenGuard |
|------|-------|---|------|----|----|----|----|----|------------|
| C10 | ASR | BadNets | 100% | 3.0% | 0.5% | 6.7% | 99.7% | 3.1% | 5.3% |
| | | Blend | 100% | 10.2% | 0.5% | 3.8% | 100% | 15.2% | **0.4%** |
| | | CLB | 100% | 1.2% | 4.0% | 21.7% | 88.3% | 0.1% | 1.0% |
| | | SIG | 94.2% | 0.4% | 0.3% | 6.8% | 89.1% | 0.01% | 2.8% |
| | | WaNet | 92.3% | 15.1% | 1.7% | 28.9% | 90.1% | 2.3% | 8.8% |
| | | SSBA | 100% | 23.7% | 0.9% | 100% | 88.3% | 4.4% | 17.1% |
| | ACC | BadNets | 93.7% | 87.2% | 90.2% | 90.1% | 92.3% | 89.1% | **92.6%** |
| | | Blend | 94.8% | 88.9% | 93.4% | 93.3% | 93.1% | 88.7% | **93.5%** |
| | | CLB | 93.8% | 91.9% | 92.7% | 91.8% | 92.4% | 89.3% | **93.3%** |
| | | SIG | 93.6% | 91.6% | 93.4% | 92.1% | 91.9% | 89.0% | 92.9% |
| | | WaNet | 92.7% | 93.5% | 90.5% | 92.2% | 87.4% | 88.5% | 92.7% |
| | | SSBA | 92.8% | 89.7% | 88.7% | 92.3% | 71.5% | 85.6% | 93.1% |
| C100 | ASR | BadNets | 99.9% | 8.9% | 6.6% | - | - | 8.8% | 7.8% |
| | | Blend | 100% | 78.1% | 3.1% | - | - | 0.5% | **0.4%** |
| | | SIG | 87.5% | 78.6% | 55.5% | - | - | 2.1% | **0.4%** |
| | | CLB | 100% | 5.1% | 6.7% | - | - | 4.3% | 3.8% |
| | ACC | BadNets | 71.8% | 68.2% | 69.7% | - | - | 66.8% | **71.5%** |
| | | Blend | 73.7% | 66.8% | 67.4% | - | - | 62.3% | **74.8%** |
| | | SIG | 74.5% | 60.7% | 63.1% | - | - | 65.3% | **74.9%** |
| | | CLB | 74.6% | 62.1% | 64.7% | - | - | 66.2% | 74.5% |
| GTSRB | ASR | BadNets | 100% | 0.5% | 0.0% | - | - | 1.0% | 2.6% |
| | | Blend | 100% | 91% | 20.7% | - | - | 23.3% | **13.7%** |
| | | SIG | 93.8% | 100% | 100% | - | - | 6.2% | 4.6% |
| | | CLB | 98.8% | 65.7% | 16.7% | - | - | 7.3% | 5.8% |
| | ACC | BadNets | 96.1% | 96.7% | 95.3% | - | - | 94.7% | 95.5% |
| | | Blend | 93.4% | 96.8% | 93.1% | - | - | 93.1% | 94.7% |
| | | SIG | 95.2% | 95.1% | 95.2% | - | - | 94.8% | **95.5%** |
| | | CLB | 95.3% | 95.4% | 94.3% | - | - | 94.3% | 95.1% |

Table 3: The attack success rate (ASR %) and the natural accuracy (ACC %) of backdoor defense methods against widely used backdoor attacks. The bold numbers mean ours is the best against the other three defense methods. We only add experiment results for CIFAR-10 due to the time limits.

Li et al. (2021a), and Spectral Signature (SS) Tran et al. (2018). In scenarios where FT and ANP necessitate additional clean datasets, we assume that defenders have access to 1% clean training data, comprising 500 images. We set the pruning threshold for ANP to 0.2 in accordance with their original configurations. Regarding the ABL method, we set the isolate rate as 10% to suit the poison rate setting of the training data. Furthermore, for consistency across all methods, we establish a batch size of 256, initiate training with a learning rate of 0.1, and employ a momentum of 0.9 over a total of 100 epochs. Additionally, we apply standard data augmentation techniques such as random cropping and horizontal flipping, consistent with their original implementations.

**Evaluation metrics.** Like former works (Wu & Wang, 2021; Li et al., 2021b), we employ two key performance metrics to evaluate various backdoor defense strategies: Attack success rate (ASR), which represents classification accuracy on the test sets with triggers. natural accuracy (ACC), which denotes the classification accuracy on clean test sets. These two metrics together offer a comprehensive view of the defense strategy's capabilities, capturing both its ability to resist backdoor attacks and its performance under natural conditions.

## 4.2 EFFECTIVENESS OF EIGENGUARD

**Comparison with existing defenses.** Table 3 provides a comprehensive illustration of the remarkable efficacy of our proposed EigenGuard across the CIFAR-10, CIFAR-100, and GTSRB datasets. In this evaluation, we assess the performance of our EigenGuard against four prevalent backdoor attack scenarios, comparing with three state-of-the-art backdoor defense techniques. The results show that our EigenGuard can outperform the other methods in most cases especially when comparing the natural accuracy on different datasets, even in situations where we lack prior knowledge of the clean data. Moreover, we need to point out that the superior performance of our EigenGuard on

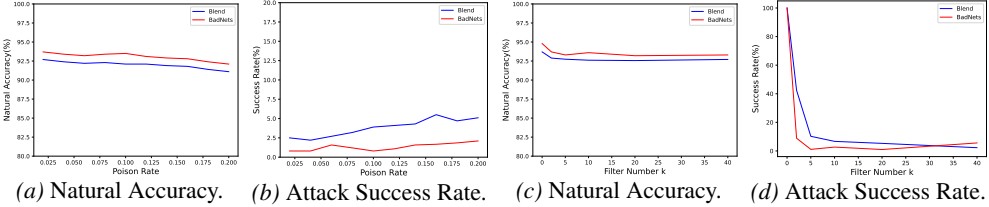

(a) Natural Accuracy.  (b) Attack Success Rate.  (c) Natural Accuracy.  (d) Attack Success Rate.

Figure 3: Natural accuracy and attack success rate for our method against different poison methods on CIFAR-10 with respect to different poison rates ((a) and (b)) and different filtering numbers $k$ for our EigenGuard ((c) and (d)).

CIFAR-100 also demonstrates that our EigenGuard can consistently preserve the model's accuracy on clean images while achieving first-class defense performance.

**Performance of our EigenGuard on different models.** In addition to assessing the performance of our EigenGuard on ResNet-18, we further evaluate its effectiveness on ResNet-34, VGG-16, and MobileNetV2, as detailed in Table 4. The results unequivocally demonstrate that our EigenGuard stands as a universal module capable of effectively safeguarding against a range of distinct backdoor attacks for various models.

|     |         | ResNet-18 | ResNet-34 | VGG-16 | MobileNetV2 |
|-----|---------|-----------|-----------|--------|-------------|
| ASR | BadNets | 5.3%      | 2.1%      | 0.6%   | 0.5%        |
|     | Blend   | 0.4%      | 3.8%      | 3.4%   | 3.3%        |
|     | CL      | 1.0%      | 1.2%      | 1.0%   | 0.4%        |
|     | SIG     | 2.8%      | 0.4%      | 0.7%   | 0.6%        |
| ACC | BadNets | 92.6%     | 93.1%     | 93.5%  | 92.1%       |
|     | Blend   | 91.1%     | 92.1%     | 92.4%  | 91.7%       |
|     | CL      | 93.3%     | 93.7%     | 94.1%  | 92.1%       |
|     | SIG     | 92.9%     | 94.1%     | 94.6%  | 94.3%       |

Table 4: The attack success rate (ASR %) and the natural accuracy (ACC %) of different models with our EigenGuard against 4 widely used backdoor attacks on CIFAR-10.

**Universality of our EigenGuard against backdoor attacks with different poison rates.** In addition to the conventional poisoning scenario which involves 10% poisoned samples in the training dataset, we conducted a comprehensive evaluation of the robustness of our EigenGuard across a range of different poison rates, spanning from 2% to 20%. The experiments are finished on CIFAR-10 with ResNet-18 and the results of these experiments are drawn in Figure 3 (a) and (b). Upon analyzing the figures, it becomes evident that as the poisoning rate increases, there is only a marginal fluctuation observed in both natural accuracy and attack success rate. These observations collectively affirm the consistency and robust defense capabilities of our EigenGuard in the face of varying poisoning rates with satisfying natural accuracy.

## 4.3 ABLATION STUDIES

Apart from the above evaluations on our proposed EigenGuard against different backdoor attacks in different scenarios, we also finish some experiments to further understand our proposed module.

**Performance of EigenGuard with different filtering numbers.** To begin our analysis, we focus on the pivotal hyperparameter, denoted as "k", within our EigenGuard framework when applied to CIFAR-10. The results of this analysis are visually depicted in Figure 3 (c) and (d). From the figure, one can see that as the value of k increases, the attack success rate of ResNet-18 equipped with our EigenGuard progressively diminishes and almost becomes zero. Consequently, we set $k = 20$ as the optimal setting for our EigenGuard framework, as detailed in our paper.

**Performance of EigenGuard attached on different deep layers.**

In our previous configurations, we strategically positioned EigenGuard after the first residual stage (before the initial down-sampling module) to minimize the influence of backdoor attacks. In this

| Location | BadNets | | Blend | |
|---|---|---|---|---|
| | ACC | ASR | ACC | ASR |
| After 1st residual stage | 92.6% | 5.3% | 93.5% | 0.4% |
| After 2nd residual stage | 92.5% | 6.2% | 93.7% | 68.4% |
| After 3rd residual stage | 93.0% | 43.2% | 93.5% | 97.9% |
| After 4th residual stage | 92.4% | 94.9% | 93.6% | 99.9% |

Table 5: The attack success rate (ASR %) and the natural accuracy (ACC %) when applying our EigenGuard after different location.

section, we delve into an examination of the impact of EigenGuard module placement within the network architecture. The results of this investigation are presented in Table 5. The figures indicate a clear trend: when EigenGuard is attached to deeper layers, the defense performance deteriorates and, in some cases, even fails to effectively mitigate backdoor attacks. One plausible explanation for this phenomenon is that the backdoor features have already dominated the semantic features for deep layers and their dimension also increases through convolution layers and skip connections. Based on this finding, we integrate our EigenGuard before the first down-sampling stage in every neural network, as detailed in our paper.

**t-SNE map for models with our EigenGuard.** In addition to the aforementioned experiments, we conducted a visual analysis of ResNet-18 features with the integration of our EigenGuard module located after different layers, as illustrated in Figure 4. This visual representation underscores a noteworthy observation: when attaching our EigenGuard, the backdoor features (depicted as "pink" points) exhibit significant overlap with samples belonging to different classes. This overlap means the model will treat the backdoor samples as natural ones. Therefore, the network will not classify these features to the target class and the network will not suffer from the backdoor threats. Furthermore, the visualization reveals that the features of distinct natural classes are distinctly separated, facilitating accurate classification. Consequently, our model not only offers robust protection against backdoor attacks but also excels in correctly predicting outcomes for natural samples.

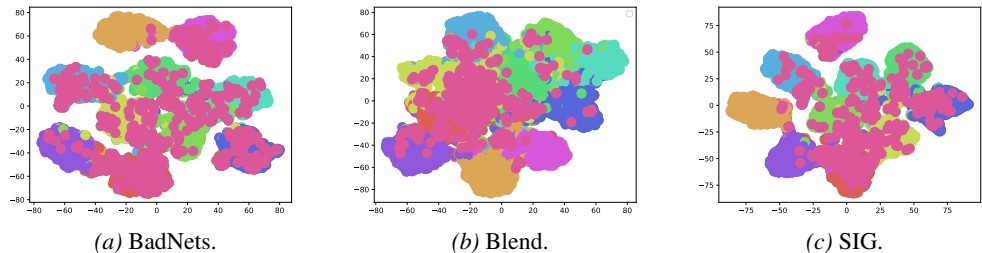

*(a)* BadNets.        *(b)* Blend.        *(c)* SIG.

Figure 4: t-SNE map on output features of ResNet-18 with our EigenGuard with respect to different eigenspace for backdoor and clean images for different datasets. The "pink" dots represent the backdoor samples while the other colors stand for different natural samples with different classes.

## 5 CONCLUSION

To tackle the challenge of training on unauthorized data, we first analyze the model's backdoor and natural behaviors from the spectral view and find that lifting the dimension of the feature space can prevent the network from making target predictions when encountering the triggers while preserving its natural performance. Building upon this insight, we introduce a novel module named Eigen-Guard. By integrating EigenGuard before the first down-sampling operation into the neural network, the model can prevent the backdoor behavior on triggers while greatly maintaining the performance of the model on natural data through natural training on the unauthorized datasets compared with other defending methods. Empirical experiments demonstrate the efficacy of our approach, indicating that models incorporating our novel modules exhibit superior performance compared to previous methodologies. This heightened performance is evident across both backdoor and natural data scenarios, surpassing the capabilities of alternative defense algorithms.

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

# A APPENDIX

## A.1 EXPERIMENT DETAILS ON BACKDOOR ATTACKS

The implementation of backdoor attacks is based on Wu & Wang (2021)'s setting. we take class $0$ as target for both CIFAR-10, CIFAR-100 and GTSRB. The details are listed in the following:

- BadNets Gu et al. (2019): The trigger is a $3 \times 3$ checkerboard at the bottom right corner of images. Given the target label, we attach the trigger to $10\%$ of training samples from other classes and relabel them as the target label.

- Blend attack Chen et al. (2017): We use a trigger pattern generated by Wu & Wang (2021) where each pixel value is sampled from a uniform distribution in [0, 255]. Given the target class, we randomly select $10\%$ of training samples from other classes for poisoning. We attach the trigger $t$ to the sample $x$ using a blended injection strategy, i.e., $\alpha t + (1\alpha)x$ with $\alpha = 0.2$. Then, we relabel them as the target label.

- Sinusoidal signal attack (SIG) Barni et al. (2019): We superimpose a sinusoidal signal over the inputs as the trigger following Barni et al. Barni et al. (2019). And the poison rate here is $80\%$.

- Clean-label Attack (CLB) Turner et al. (2019): The trigger is a $3 \times 3$ checkerboard at the four corners of images and the poisoning rate is $80\%$. To make the poisoning process much easier, we apply adversarial perturbations to render these poisoned samples harder to classify during training following Turner et al. (2019)'s settings. Specifically, we use Projected Gradient Descent (PGD) to generate adversarial perturbations with perturbation budgets equal to $16/255$.

## A.2 PROOF OF THE PROPOSITION

*Proof.* If the model is going to predict the input $\mathbf{x}$ as the natural class instead of the target classes, the final output should be positive, which means:

$$\mathbf{w}^\top \mathbf{z} > 0 \tag{4}$$

$$\mathbf{w}^\top \sum_i^5 \alpha_i \mathbf{f}_i > 0 \tag{5}$$

$$\mathbf{w}^\top \sum_i^4 \alpha_i \mathbf{f}_i > -\mathbf{w}^\top \alpha_5 \mathbf{f}_5 \tag{6}$$

$\square$

Since $\mathbf{w}^\top \mathbf{f}_i \in (\beta, 1)$ if $i = 1, 2, 3, 4$, $\mathbf{w}^\top \mathbf{f}_5 \in (-1, -\beta)$, the worst case is when all $\mathbf{w}^\top \mathbf{f}_i = \beta$ for $i = 1, 2, 3, 4$ and $\mathbf{w}^\top \mathbf{f}_5 = -1$. Then we need to find when the following inequality holds:

$$\beta \sum_i^4 \alpha_i > \alpha_5 = (1 - \sum_i^4 \alpha_i), \tag{7}$$

Then we can get the above inequality holds only if the following condition holds:

$$\sum_i^4 \alpha_i > \frac{1}{1+\beta}. \tag{8}$$

Then we have the following inequality:

$$\mathbf{w}^\top \sum_i^4 \alpha_i \mathbf{f}_i > \beta \sum_i^4 \alpha_i > \alpha_5 > -\mathbf{w}^\top \alpha_5 \mathbf{f}_5. \tag{9}$$

Therefore, the model will predict the natural class instead of the target class.

