# OpenReview forum: "EigenGuard: Backdoor Defense in Eigenspace"
_ICLR.cc/2024/Conference — Submitted to ICLR 2024_

### Official Review · Reviewer_5Wz3 · 2023-10-26

**Soundness:** 3 good
**Presentation:** 2 fair
**Contribution:** 2 fair
**Rating:** 5
**Confidence:** 4

**Summary:**

In this paper, the authors observe that trigger features, often referred to as backdoor features, present a distinct concentrated behavior within the spectral space, especially around the top singular values. Based on these insights, they introduce "EigenGuard", a method designed to mitigate the impact of trigger features by scaling the top k spectral features during the training process. Experiments show that, models with EigenGuard show better performance on both backdoor and natural examples compared with other defense algorithms.

**Strengths:**

* The authors propose a method that does not require extra clean data and an unlearning process but can still remove the impact of triggers and enable training clean models on untrusted datasets.

* The proposed method surpasses the previous works on some attacks.

* The paper provides an ablation study to investigate the effect of the singular value $k$ and layers.

**Weaknesses:**

* The findings are not novel. SM Moosavi-Dezfooli et al.[Ref-1] present that universal perturbations exhibit a concentrated behavior within the spectral space. Backdoor triggers, as a special type of universal perturbation, exhibit a similar property, which appears to lack significance.

[Ref-1] Moosavi-Dezfooli, Seyed-Mohsen, et al. "Universal adversarial perturbations." Proceedings of the IEEE conference on computer vision and pattern recognition. 2017.

* Some arguments lack validation. For instance, the authors claim that existing detection-unlearning  based mitigations may induce a decrease in accuracy as the neural network may forget many useful features for classifying clean samples. However, in this paper, the authors do not support this argument with references or experiments. Contradictorily, as shown in Tab. 3, fine-tuning maintains better ACC than EigenGuard. How do these experimental results support the aforementioned argument? In addition, I do not believe that modifying the dimension of the feature space does not degrade the prediction accuracy for clean or natural images. The authors should justify why their manipulation of the dimension of feature space does not lead to ACC drops.

* Design choices are unclear.
    * After progressively eliminating the dimension of the feature space by setting the top singular values of features SVD decomposition to 0, why is it necessary to lift the original small singular values and generate new features? Doesn't this manipulation lead to a degradation of the prediction accuracy for clean or natural images?
    * What is the rationale for scaling $\sigma_k$ by 0.001 in Algorithm 1, and how is this scale factor determined?

* The paper lacks theoretical proof of the proposed method.

* Writing needs improvement.

* The evaluation lacks comprehensiveness and does not include comparisons with some related works:

   * For defenses:

       * Zhenting Wang, Kai Mei, Hailun Ding, Juan Zhai, and  Shiqing Ma. Rethinking the reverse-engineering of trojan triggers. In Advances in Neural Information Processing Systems, 2022.

      * Jonathan Hayase and Weihao Kong. Spectre: Defending  against backdoor attacks using robust covariance estimation.  In International Conference on Machine Learning, 2020

    * For attacks:

      * Siyuan Cheng, Yingqi Liu, Shiqing Ma, and Xiangyu Zhang. Deep feature space trojan attack of neural networks by controlled detoxification. In AAAI, 2021

      * Tuan Anh Nguyen and Anh Tran. Input-aware dynamic backdoor attack. Advances in Neural Information Processing Systems, 33:3454–3464, 2020

      * Li, Shaofeng, et al. "Invisible backdoor attacks on deep neural networks via steganography and regularization." IEEE Transactions on Dependable and Secure Computing 18.5 (2020): 2088-2105.

      * Nguyen, Anh, and Anh Tran. "Wanet--imperceptible warping-based backdoor attack." arXiv preprint arXiv:2102.10369 (2021).

**Questions:**

* Is the proposed method able to purify more complex attacks? (See weakness missing literatures)

* What could be an adaptive attack and will the proposed method still be effective?

---

> ### Author Response · Authors · 2023-11-20
> **Rebuttal Part 1**
>
> **1. About the difference between our paper and Universal Perturbations:**
>
> Although trigger features can be regarded as a kind of universal perturbation that lies in eigenspace corresponding to the top singular value, the former work doesn’t show that trigger features are low rank and can only lie in top eigenspace. Furthermore, the relationship between the feature dimensions and backdoor accuracy during training as shown in Figure 1 is only unevaluatedin our paper. Therefore, the new findings of our paper are that the trigger features can only located in the top low-rank subspaces. Such a phenomenon may be caused by the low poison rate of the trigger samples.
>
> Furthermore, we also designed our new method by damping top singular values and using features corresponding to more eigenspace for prediction. Therefore, models with our module can naturally defend against many backdoor attacks.
>
> **2. About “lack of evidence that the unlearning will lead to bad natural accuracy” and “ finetuning is better than EigenGuard”:**
>
> From Table 3, one can see that ABL’s natural performance is not good, which is caused by its unlearning loss.
>
> As for finetuning, its natural accuracy may be better sometimes because they use additional clean datasets for finetuning while ours do not. But its backdoor accuracy is not stable which means that fine-tuning is not an effective way for backdoor defense.
>
> **3. About why modifying eigenspace will not cause the accuracy drops:**
>
> Because we do not drop features, we only rescale the singular values corresponding to each eigenspace.  As our experiments show, the trigger features only lie in a low-rank space with large singular values, our method will lead neural networks to make predictions based on more features instead of trigger features. Therefore, natural features will help the neural network to predict natural labels instead of trigger labels on backdoor samples.
>
> As for natural samples, one can notice that models with our EigenGuard do not drop any information. Therefore, the natural performance will not be affected.
>
> **4. Why not just let top singular values to zero?**
>
> Just removing the top eigenspace will make the natural accuracy drop because features corresponding to top singular values are also important for natural classification, as shown in the following experiments:
>
> | Methods | ACC. | ASR. |
> | -------- | ------- | ------- |
> | Remove top 10 Singulars  |  $85.4\%$   |  $4.2\%$   |
> | Ours | $92.6\%$     |  $5.3\%$   |
>
>
> Our method tries to avoid trigger features dominating the prediction. According to our theory, if the trigger features not dominating, which means the corresponding singular value is not large, the rest natural features will help models predict other classes instead of natural classes.
>
> **5. About the 0.001 scaling factor.**
>
> This parameter will not influence much, we don’t set it to zero because we think the gradient corresponds to the latter eigenspace may also help. But in fact, such a hyper-parameter will not influence much.
>
> **6. About the theoretical proof.**
>
> Our theory illustrates that damping the dominating trigger features may make the models use natural features to make correct predictions. It is also consistent with our method, we make the top singular values smaller to avoid the triggers dominating classifications.
>
> **7. About more baselines:**
>
> As for baselines, we choose the most effective attacks and defenses as baselines according to BackdoorBench, since we use its setting for evaluation. We also add some defense methods(NAD[1], Spectral signature (SS)[2]) and attack methods (WaNet[3], SSBA[4]) on CIFAR-10. The full table is shown below:

---

> > ### Author Response · Authors · 2023-11-20
> > **Rebuttal Part 2**
> >
> > |                     | | Types             | None                 | FT                     | ANP                   | ***NAD***                    | ***SS***                    | ABL                   | EigenGuard            |
> > |---------------------|-------------------|-------------------|----------------------|------------------------|-----------------------|------------------------|-----------------------|-----------------------|-----------------------|
> > | C10                 | ASR               | BadNets              | $100\%$                | $3.0\%$               | $0.5\%$                | $6.7\%$  | $99.7\%$ | $3.1\%$  | $5.3\%$             |
> > |                     |                   | Blend                | $100\%$                | $10.2\%$              | $0.5\%$                | $3.8\%$  | $100\%$  | $15.2\%$ | $0.4\%$             |
> > |                     |                   | CLB                  | $100\%$                | $1.2\%$               | $4.0\%$                | $21.7\%$ | $88.3\%$ | $0.1\%$ | $1.0\%$              |
> > |                     |                   | SIG                  | $94.2\%$               | $0.4\%$               | $0.3\%$                | $6.8\%$  | $89.1\%$ | $0.01\%$   | $2.8\%$           |
> > |                     |                   | ***WaNet***                | $92.3\%$  | $15.1\%$ | $1.7\%$   | $28.9\%$ | $90.1\%$ | $2.3\%$  | $8.8\%$|
> > |                     |                   | ***SSBA***                 | $100\%$   | $23.7\%$ | $0.9\%$   | $100\%$  | $88.3\%$ | $4.4\%$  | $17.1\%$|
> > |                     | ACC               | BadNets              | $93.7\%$               | $87.2\%$              | $90.2\%$        x       | $90.1\%$ | $92.3\%$ | $89.1\%$  | $92.6\%$             |
> > |                     |                   | Blend                | $94.8\%$               | $88.9\%$              | $93.4\%$               | $93.3\%$ | $93.1\%$ | $88.7\%$  | $93.5\%$            |
> > |                     |                   | CLB                  | $93.8\%$               | $91.9\%$              | $92.7\%$               | $91.8\%$ | $92.4\%$ | $89.3\%$  | $93.3\%$            |
> > |                     |                   | SIG                  | $93.6\%$               | $91.6\%$              | $93.4\%$               | $92.1\%$ | $91.9\%$ | $89.0\%$  | $92.9\%$            |
> > |                     |                   | ***WaNet***                | $92.7\%$  | $93.5\%$ | $90.5\%$  | $92.2\%$ | $87.4\%$ | $88.5\%$ | $92.7\%$|
> > |                     |                   | ***SSBA***   | $92.8\%$  | $89.7\%$ | $88.7\%$  | $92.3\%$ | $71.5\%$ | $85.6\%$ | $93.1\%$|
> > | C100  | ASR             | BadNets              | $99.9\%$               | $8.9\%$               | $6.6\%$                | -        | -        | $8.8\%$               | $7.8\%$ |
> > |                     |                   | Blend                | $100\%$                | $78.1\%$              | $3.1\%$                | -        | -        | $0.5\%$               |  $0.4\%$ |
> > |                     |                   | SIG                  | $87.5\%$               | $78.6\%$              | $55.5\%$               | -        | -        | $2.1\%$               |  $0.4\%$ |
> > |                     |                   | ***CLB***     | $100\%$   | $5.1\%$  | $6.7\%$   | -        | -        | $4.3\%$  | $3.8\%$|
> > |                     | ACC | BadNets              | $71.8\%$               | $68.2\%$              | $69.7\%$               | -        | -        | $66.8\%$              | $74.1\%$ |
> > |                     |                   | Blend                | $73.7\%$               | $66.8\%$              | $67.4\%$               | -        | -        | $62.3\%$              | $74.8\%$ |
> > |                     |                   | SIG                  | $74.5\%$               | $60.7\%$              | $63.1\%$               | -        | -        | $65.3\%$              | $74.9\%$ |
> > |                     |                   | ***CLB***     | $74.6\%$  | $62.1\%$ | $64.7\%$  | -        | -        | $66.2\%$ | $74.5\%$ |

---

> > > ### Author Response · Authors · 2023-11-20
> > > **Rebuttal Part 3**
> > >
> > > |                     | | Types             | None                 | FT                     | ANP                   | ***NAD***                    | ***SS***                    | ABL                   | EigenGuard            |
> > > |---------------------|-------------------|-------------------|----------------------|------------------------|-----------------------|------------------------|-----------------------|-----------------------|-----------------------|
> > > | GTSRB | ASR | BadNets              | $100\%$                | $0.5\%$               | $0.0\%$                | -        | -        | $1.0\%$               | $2.6\%$ |
> > > |                     |                   | Blend                | $100\%$                | $91\%$                | $20.7\%$               | -        | -        | $23.3\%$            | $13.7\%$ |
> > > |                     |                   | SIG                  | $93.8\%$               | $100\%$               | $100\%$                | -        | -        | $6.2\%$               | $4.6\%$ |
> > > |                     |                   | ***CLB***     | $98.8\%$  | $65.7\%$ | $16.7\%$  | -        | -        | $7.3\%$  | $5.8\%$ |
> > > |                     | ACC | BadNets              | $96.1\%$               | $96.7\%$              | $95.3\%$               | -        | -        | $94.7\%$              | $95.5\%$ |
> > > |                     |                   | Blend                | $93.4\%$               | $96.8\%$              | $93.1\%$               | -        | -        | $93.1\%$              | $94.7\%$ |
> > > |                     |                   | SIG                  | $95.2\%$               | $95.1\%$              | $95.2\%$               | -        | -        | $94.8\%$              | $95.5\%$ |
> > > |                     |                   | ***CLB***     | $95.3\%$  | $95.4\%$ | $94.3\%$  | -        | -        | $94.3\%$ | $95.1\%$ |
> > >
> > > [1] Neural Attention Distillation: Erasing Backdoor Triggers From Deep Neural Networks.
> > >
> > > [2] Spectral signatures in backdoor attacks.
> > >
> > > [3] WaNet -- Imperceptible Warping-Based Backdoor Attack
> > >
> > > [4] Invisible Backdoor Attack with Sample-Specific Triggers
> > >
> > > The results show that our method can perform well on different attacks. Apart from that, we also noticed that the spectral signature cannot perform well under the backdoor bench’s setting.
> > >
> > > Apart from that, we also noticed that the spectral signature cannot perform well under the backdoor bench’s setting. The phenomenon may caused by its dataset split being parametric sensitive and the undetected few backdoor samples that are not detected can also lead to backdoor behavior in backdoor training, which also demonstrates that the detection and retraining defense methods are unstable.

---

### Official Review · Reviewer_Zcuo · 2023-10-26

**Soundness:** 2 fair
**Presentation:** 3 good
**Contribution:** 3 good
**Rating:** 5
**Confidence:** 2

**Summary:**

The paper addresses the vulnerability of Deep Neural Networks (DNNs) to backdoor attacks from poisoned training data. It introduces a novel module called EigenGuard, which helps DNNs neglect backdoor triggers while maintaining performance on legitimate data. Through experiments, the authors show that models equipped with EigenGuard outperform other defense algorithms in handling both poisoned and clean data. This work contributes to enhancing the security of DNNs against backdoor attacks in scenarios with potentially unreliable training datasets.

**Strengths:**

- The idea of this seems to be novel. The authors find that forcing a high-dimensional feature space will make backdoor images fail to attack.
- This paper provides a theoretical understanding of the proposed method.
- The paper is easy to follow.
- The experimental results seem to be good. It outperforms other defense methods.

**Weaknesses:**

- The paper could be strengthened by addressing the potential of adaptive attacks, especially when attackers know the EigenGuard defense.
- Why is CLR only evaluated on CIFAR10? The authors should justify it. Also, for the defense choices, the authors should justify why these methods are considered. As far as I know, many other SOTA defenses are not considered [1,2,5].
- The paper would benefit from discussing the effectiveness of EigenGuard against self-supervised learning backdoor attacks [3-5], as the proposed defense operates on the output of the encoder (i.e., f). This exploration could significantly enhance the robustness and applicability of the proposed method.
- To bolster the generalizability of the findings, it would be advantageous to evaluate the effectiveness of EigenGuard across a variety of model architectures.
- Minor issues, such as the newline problem in Section 4.3, should be rectified for improved readability and professionalism.

[1] Chen, Bryant, et al. "Detecting backdoor attacks on deep neural networks by activation clustering." arXiv preprint arXiv:1811.03728 (2018).

[2] Tang, Di, et al. "Demon in the variant: Statistical analysis of {DNNs} for robust backdoor contamination detection." 30th USENIX Security Symposium (USENIX Security 21). 2021.

[3] Saha A, Tejankar A, Koohpayegani S A, et al. Backdoor attacks on self-supervised learning[C]//Proceedings of the IEEE/CVF Conference on Computer Vision and Pattern Recognition. 2022: 13337-13346.

[4] Li, Changjiang, et al. "Demystifying Self-supervised Trojan Attacks." arXiv preprint arXiv:2210.07346 (2022).

[5] Feng, Shiwei, et al. "Detecting Backdoors in Pre-trained Encoders." Proceedings of the IEEE/CVF Conference on Computer Vision and Pattern Recognition. 2023.

**Questions:**

Please see the weakness.

---

> ### Author Response · Authors · 2023-11-20
> **Rebuttal Part 1**
>
> Thanks for your review, the following are our responses.
>
> **1. About the possible adaptive attacks:**
>
> A possible adaptive attack is trying to expand the trigger feature space and make it as complicated as the natural features. Then only damping top eigenspaces for features is not useful because trigger features are varied enough and no natural features are used for classification.
>
> However, we think it is not easy to achieve because too complicated backdoor features will also make the backdoor not easy to implement. Because the success of backdoor attacks benefits from the simple shortcuts between trigger features and labels. Therefore, the attack may be less effective when trigger features are complicated.
>
> **2. About CLB attacks and the choice of baselines.**
>
> We also evaluate the CLB attacks on CIFAR-100 and GTSRB attacks. And the results show that our methods are also effective. As for the choice of baselines, we choose the most effective attacks and defenses as baselines according to BackdoorBench, since we use its setting for evaluation. We also add some defense methods(NAD[1], Spectral Signature (SS[2]) defense ) and attack methods (WaNet[3],SSBA[4]) on CIFAR-10. The full table is shown below:
>
> |                     | | Types             | None                 | FT                     | ANP                   | ***NAD***                    | ***SS***                    | ABL                   | EigenGuard            |
> |---------------------|-------------------|-------------------|----------------------|------------------------|-----------------------|------------------------|-----------------------|-----------------------|-----------------------|
> | C10                 | ASR               | BadNets              | $100\%$                | $3.0\%$               | $0.5\%$                | $6.7\%$  | $99.7\%$ | $3.1\%$  | $5.3\%$             |
> |                     |                   | Blend                | $100\%$                | $10.2\%$              | $0.5\%$                | $3.8\%$  | $100\%$  | $15.2\%$ | $0.4\%$             |
> |                     |                   | CLB                  | $100\%$                | $1.2\%$               | $4.0\%$                | $21.7\%$ | $88.3\%$ | $0.1\%$ | $1.0\%$              |
> |                     |                   | SIG                  | $94.2\%$               | $0.4\%$               | $0.3\%$                | $6.8\%$  | $89.1\%$ | $0.01\%$   | $2.8\%$           |
> |                     |                   | ***WaNet***                | $92.3\%$  | $15.1\%$ | $1.7\%$   | $28.9\%$ | $90.1\%$ | $2.3\%$  | $8.8\%$|
> |                     |                   | ***SSBA***                 | $100\%$   | $23.7\%$ | $0.9\%$   | $100\%$  | $88.3\%$ | $4.4\%$  | $17.1\%$|
> |                     | ACC               | BadNets              | $93.7\%$               | $87.2\%$              | $90.2\%$        x       | $90.1\%$ | $92.3\%$ | $89.1\%$  | $92.6\%$             |
> |                     |                   | Blend                | $94.8\%$               | $88.9\%$              | $93.4\%$               | $93.3\%$ | $93.1\%$ | $88.7\%$  | $93.5\%$            |
> |                     |                   | CLB                  | $93.8\%$               | $91.9\%$              | $92.7\%$               | $91.8\%$ | $92.4\%$ | $89.3\%$  | $93.3\%$            |
> |                     |                   | SIG                  | $93.6\%$               | $91.6\%$              | $93.4\%$               | $92.1\%$ | $91.9\%$ | $89.0\%$  | $92.9\%$            |
> |                     |                   | ***WaNet***                | $92.7\%$  | $93.5\%$ | $90.5\%$  | $92.2\%$ | $87.4\%$ | $88.5\%$ | $92.7\%$|
> |                     |                   | ***SSBA***   | $92.8\%$  | $89.7\%$ | $88.7\%$  | $92.3\%$ | $71.5\%$ | $85.6\%$ | $93.1\%$|

---

> > ### Author Response · Authors · 2023-11-20
> > **Rebuttal Part 2**
> >
> > |                     | | Types             | None                 | FT                     | ANP                   | NAD                    | SS                    | ABL                   | EigenGuard            |
> > |---------------------|-------------------|-------------------|----------------------|------------------------|-----------------------|------------------------|-----------------------|-----------------------|-----------------------|
> > | C100  | ASR             | BadNets              | $99.9\%$               | $8.9\%$               | $6.6\%$                | -        | -        | $8.8\%$               | $7.8\%$ |
> > |                     |                   | Blend                | $100\%$                | $78.1\%$              | $3.1\%$                | -        | -        | $0.5\%$               |  $0.4\%$ |
> > |                     |                   | SIG                  | $87.5\%$               | $78.6\%$              | $55.5\%$               | -        | -        | $2.1\%$               |  $0.4\%$ |
> > |                     |                   | ***CLB***     | $100\%$   | $5.1\%$  | $6.7\%$   | -        | -        | $4.3\%$  | $3.8\%$|
> > |                     | ACC | BadNets              | $71.8\%$               | $68.2\%$              | $69.7\%$               | -        | -        | $66.8\%$              | $74.1\%$ |
> > |                     |                   | Blend                | $73.7\%$               | $66.8\%$              | $67.4\%$               | -        | -        | $62.3\%$              | $74.8\%$ |
> > |                     |                   | SIG                  | $74.5\%$               | $60.7\%$              | $63.1\%$               | -        | -        | $65.3\%$              | $74.9\%$ |
> > |                     |                   | ***CLB***     | $74.6\%$  | $62.1\%$ | $64.7\%$  | -        | -        | $66.2\%$ | $74.5\%$ |
> > | GTSRB | ASR | BadNets              | $100\%$                | $0.5\%$               | $0.0\%$                | -        | -        | $1.0\%$               | $2.6\%$ |
> > |                     |                   | Blend                | $100\%$                | $91\%$                | $20.7\%$               | -        | -        | $23.3\%$            | $13.7\%$ |
> > |                     |                   | SIG                  | $93.8\%$               | $100\%$               | $100\%$                | -        | -        | $6.2\%$               | $4.6\%$ |
> > |                     |                   | ***CLB***     | $98.8\%$  | $65.7\%$ | $16.7\%$  | -        | -        | $7.3\%$  | $5.8\%$ |
> > |                     | ACC | BadNets              | $96.1\%$               | $96.7\%$              | $95.3\%$               | -        | -        | $94.7\%$              | $95.5\%$ |
> > |                     |                   | Blend                | $93.4\%$               | $96.8\%$              | $93.1\%$               | -        | -        | $93.1\%$              | $94.7\%$ |
> > |                     |                   | SIG                  | $95.2\%$               | $95.1\%$              | $95.2\%$               | -        | -        | $94.8\%$              | $95.5\%$ |
> > |                     |                   | ***CLB***     | $95.3\%$  | $95.4\%$ | $94.3\%$  | -        | -        | $94.3\%$ | $95.1\%$ |
> >
> > [1] Neural Attention Distillation: Erasing Backdoor Triggers From Deep Neural Networks.
> >
> > [2] Spectral signatures in backdoor attacks.
> >
> > [3] WaNet -- Imperceptible Warping-Based Backdoor Attack
> >
> > [4] Invisible Backdoor Attack with Sample-Specific Triggers
> >
> > The results show that our method can perform well on different attacks. Apart from that, we also noticed that the spectral signature cannot perform well under the backdoor bench’s setting.
> >
> > The phenomenon may caused by its dataset split being parametric sensitive and the undetected few backdoor samples that are not detected can also lead to backdoor behavior in backdoor training, which also demonstrates that the detection and retraining defense methods are unstable.
> >
> > **3. About EigenGuard on the self-supervised learning backdoor attacks:**
> >
> > Thanks for your advice, we also evaluate the performance of CTRL on ResNet-18 with SimCLR on CIFAR-10. The results are as follows:
> >
> > | Methods | ACC. | ASR. |
> > | -------- | ------- | ------- |
> > | Ori  |  $80.5\%$   |  $85.3\%$   |
> > | EigenGuard | $80.3\%$     |  $22.1\%$   |
> >
> > From the table, one can see that our EigenGuard can also effectively reduce the backdoor performance of SSL attacks.

---

> > > ### Author Response · Authors · 2023-11-20
> > > **Rebuttal - Part3**
> > >
> > > **4. About model architecture:**
> > >
> > > Our EigenGuard can be used in many model architectures with effective performance:
> > >
> > > |  |  | ResNet-18 | ResNet-34 | VGG-16 | MobileNetV2 |
> > > | --- | --- | --- | --- | --- | --- |
> > > | ASR | BadNets | $ 5.3\%$ | $2.1\%$ | $0.6\%$ | $0.5\%$ |
> > > |  | Blend | $0.4\%$ | $3.8\%$ | $3.4\%$ | $3.3\%$ |
> > > |  | CL | $1.0\%$ | $1.2\%$ | $1.0\%$ | $0.4\%$ |
> > > |  | SIG | $2.8\%$ | $0.4\%$ | $0.7\%$ | $0.6\%$ |
> > > | ACC | BadNets | $ 92.6\%$ | $93.1\%$ | $93.5\%$ | $92.1\%$ |
> > > |  | Blend | $91.1\%$ | $92.1\%$ | $92.4\%$ | $91.7\%$ |
> > > |  | CL | $93.3\%$ | $93.7\%$ | $94.1\%$ | $92.1\%$ |
> > > |  | SIG | $92.9\%$ | $94.1\%$ | $94.6\%$ | $94.3\%$ |

---

> > > > ### Comment · Reviewer_Zcuo · 2023-11-22
> > > >
> > > > The author addressed most of my concerns. However, the adaptive attack concern is not well-addressed. As we all know, many defense papers are broken by adaptive or stronger attacks.  The authors should explore some intuitive, adaptive attacks, and provide discussion and insights to the community.

---

> > > > > ### Author Response · Authors · 2023-11-23
> > > > > **About the possible adapative attack**
> > > > >
> > > > > Another adaptive attack is trying to break the natural features, therefore model cannot make accurate predictions by utilizing natural features and may still predict triggers based on trigger features even with our EigenGuard. Therefore, we add PGD-10 adversarial noise of a **pre-trained ResNet18 with our EigenGuard** with $\epsilon=8/255$ to the poisoned class to break natural features and then do the BadNet attack on such dataset. The results are shown below:
> > > > >
> > > > > | Methods | ACC. | ASR. |
> > > > > | -------- | ------- | ------- |
> > > > > | ResNet-18 |  $84.9\%$   |  $99.4\%$   |
> > > > > | ResNet-18 with our EigenGuard | $84.1\%$     |  $5.4\%$   |
> > > > >
> > > > > The degeneration of the natural accuracy is caused by the injection of adversarial noise. From the result, one can see that our EigenGuard can still defend it **even if the attack is generated by a pre-trained ResNet-18 with EigenGuard**. Such a result demonstrates that useful natural features are much more than we expected and cannot be fully destroyed.

---

### Official Review · Reviewer_6rWf · 2023-11-01

**Soundness:** 2 fair
**Presentation:** 3 good
**Contribution:** 2 fair
**Rating:** 5
**Confidence:** 3

**Summary:**

The paper proposes a novel defense method called EigenGuard to mitigate backdoor attacks on deep neural networks. The authors first analyze the spectral behavior of features in neural networks and observe that backdoor features tend to exhibit a concentrated behavior within the spectral space, while natural features are distributed in a high-dimensional space. Based on these observations, the authors propose the EigenGuard module, which forces the top k spectral features to share the same scale during training. This module effectively neutralizes the impact of backdoor connections while preserving the natural performance of the model. Experimental results demonstrate that EigenGuard outperforms three existing defense methods in terms of both backdoor attack success rate and natural accuracy.

**Strengths:**

1. The paper proposes a novel defense method, EigenGuard, which leverages the spectral behavior of features in neural networks to mitigate backdoor attacks during model training.

2. The paper is well written and easy to follow.

**Weaknesses:**

1. The theoretical foundation of the paper seems derivative, lacking novelty.

2. The paper does not provide a comprehensive review of related works and omits comparisons with well-established baselines.

**Questions:**

I have a few concerns regarding the proposed method and experiments：

- The defense strategy hinges on the premise that backdoor features and genuine features are distributed in distinct spectral spaces. This foundational idea has already been explored by prior works such as [1, 2] for backdoor mitigation. While this paper implements the theory to devise algorithms that defense backdoor attacks during model training, the theoretical underpinning raises questions regarding its novelty.

- The paper overlooks some of the more recent developments in backdoor attacks and defenses. As a result, the experiments lack a comprehensive scope. Omissions include backdoor attacks like [3-6] and backdoor defenses during training such as those presented in [7, 8]. For a more holistic overview, the paper may refer to existing work [9].

Other questions:

- I feel the details of the threat model are missing.

- Some terms such as **$A$** in formula 1 and $k, \sigma_k$ in algorithm 1 are unclear.

- It is confusing why the CLB is not considered for experiments (e.g, Cifar100 and GTSRB).

- From Table 3, it's observed that some models, when defended with EigenGuard, exhibit improved accuracy compared to when no defense is applied. Is there a rationale behind this outcome?

[1] Tran, B., Li, J. and Madry, A., 2018. Spectral signatures in backdoor attacks. *Advances in neural information processing systems*, *31*.

[2] Karim, N., Arafat, A.A., Khalid, U., Guo, Z. and Rahnavard, N., 2023. Efficient Backdoor Removal Through Natural Gradient Fine-tuning. *arXiv preprint arXiv:2306.17441*.

[3] Wang, Z., Zhai, J. and Ma, S., 2022. Bppattack: Stealthy and efficient trojan attacks against deep neural networks via image quantization and contrastive adversarial learning. In *Proceedings of the IEEE/CVF Conference on Computer Vision and Pattern Recognition* (pp. 15074-15084).

[4] Li, Y., Li, Y., Wu, B., Li, L., He, R. and Lyu, S., 2021. Invisible backdoor attack with sample-specific triggers. In *Proceedings of the IEEE/CVF international conference on computer vision* (pp. 16463-16472).

[5] Nguyen, A. and Tran, A., 2021. Wanet--imperceptible warping-based backdoor attack. *arXiv preprint arXiv:2102.10369*.

[6] Cheng, S., Liu, Y., Ma, S. and Zhang, X., 2021, May. Deep feature space trojan attack of neural networks by controlled detoxification. In *Proceedings of the AAAI Conference on Artificial Intelligence* (Vol. 35, No. 2, pp. 1148-1156).

[7] Wang, Z., Ding, H., Zhai, J. and Ma, S., 2022. Training with more confidence: Mitigating injected and natural backdoors during training. *Advances in Neural Information Processing Systems*, *35*, pp.36396-36410.

[8] Li, Y., Lyu, X., Koren, N., Lyu, L., Li, B. and Ma, X., 2021. Neural attention distillation: Erasing backdoor triggers from deep neural networks. *arXiv preprint arXiv:2101.05930*.

[9] Li, Y., Zhang, S., Wang, W. and Song, H., 2023. Backdoor Attacks to Deep Learning Models and Countermeasures: A Survey. *IEEE Open Journal of the Computer Society*.

---

> ### Author Response · Authors · 2023-11-20
> **Rebuttal Part 1**
>
> Thanks for your review, the following are our responses.
>
> **1. About our theoretical analysis:**
>
> Although our toy model is simple, our theory can clearly illustrate that damping the dominating trigger features will lead the models to use natural features to make correct predictions. It is consistent with our method, we make the top singular values smaller to avoid the triggers dominating classifications.
>
> **2. About the baselines in our paper:**
>
> As for the choice of baselines, we choose the most effective attacks and defenses as baselines according to BackdoorBench, since we use its setting for evaluation. We also add some defense methods (NAD[1] and Spectral Signature [2]) and attack methods(WaNet[3] and SSBA[4]) on CIFAR-10. The results show that our method can perform well on different attacks. The full table is shown below:
>
> |                     | | Types             | None                 | FT                     | ANP                   | ***NAD***                    | ***SS***                    | ABL                   | EigenGuard            |
> |---------------------|-------------------|-------------------|----------------------|------------------------|-----------------------|------------------------|-----------------------|-----------------------|-----------------------|
> | C10                 | ASR               | BadNets              | $100\%$                | $3.0\%$               | $0.5\%$                | $6.7\%$  | $99.7\%$ | $3.1\%$  | $5.3\%$             |
> |                     |                   | Blend                | $100\%$                | $10.2\%$              | $0.5\%$                | $3.8\%$  | $100\%$  | $15.2\%$ | $0.4\%$             |
> |                     |                   | CLB                  | $100\%$                | $1.2\%$               | $4.0\%$                | $21.7\%$ | $88.3\%$ | $0.1\%$ | $1.0\%$              |
> |                     |                   | SIG                  | $94.2\%$               | $0.4\%$               | $0.3\%$                | $6.8\%$  | $89.1\%$ | $0.01\%$   | $2.8\%$           |
> |                     |                   | ***WaNet***                | $92.3\%$  | $15.1\%$ | $1.7\%$   | $28.9\%$ | $90.1\%$ | $2.3\%$  | $8.8\%$|
> |                     |                   | ***SSBA***                 | $100\%$   | $23.7\%$ | $0.9\%$   | $100\%$  | $88.3\%$ | $4.4\%$  | $17.1\%$|
> |                     | ACC               | BadNets              | $93.7\%$               | $87.2\%$              | $90.2\%$        x       | $90.1\%$ | $92.3\%$ | $89.1\%$  | $92.6\%$             |
> |                     |                   | Blend                | $94.8\%$               | $88.9\%$              | $93.4\%$               | $93.3\%$ | $93.1\%$ | $88.7\%$  | $93.5\%$            |
> |                     |                   | CLB                  | $93.8\%$               | $91.9\%$              | $92.7\%$               | $91.8\%$ | $92.4\%$ | $89.3\%$  | $93.3\%$            |
> |                     |                   | SIG                  | $93.6\%$               | $91.6\%$              | $93.4\%$               | $92.1\%$ | $91.9\%$ | $89.0\%$  | $92.9\%$            |
> |                     |                   | WaNet                | $92.7\%$  | $93.5\%$ | $90.5\%$  | $92.2\%$ | $87.4\%$ | $88.5\%$ | $92.7\%$|
> |                     |                   | SSBA    | $92.8\%$  | $89.7\%$ | $88.7\%$  | $92.3\%$ | $71.5\%$ | $85.6\%$ | $93.1\%$|
> | C100  | ASR             | BadNets              | $99.9\%$               | $8.9\%$               | $6.6\%$                | -        | -        | $8.8\%$               | $7.8\%$ |
> |                     |                   | Blend                | $100\%$                | $78.1\%$              | $3.1\%$                | -        | -        | $0.5\%$               |  $0.4\%$ |
> |                     |                   | SIG                  | $87.5\%$               | $78.6\%$              | $55.5\%$               | -        | -        | $2.1\%$               |  $0.4\%$ |
> |                     |                   | ***CLB***     | $100\%$   | $5.1\%$  | $6.7\%$   | -        | -        | $4.3\%$  | $3.8\%$|
> |                     | ACC | BadNets              | $71.8\%$               | $68.2\%$              | $69.7\%$               | -        | -        | $66.8\%$              | $74.1\%$ |
> |                     |                   | Blend                | $73.7\%$               | $66.8\%$              | $67.4\%$               | -        | -        | $62.3\%$              | $74.8\%$ |
> |                     |                   | SIG                  | $74.5\%$               | $60.7\%$              | $63.1\%$               | -        | -        | $65.3\%$              | $74.9\%$ |
> |                     |                   | ***CLB***     | $74.6\%$  | $62.1\%$ | $64.7\%$  | -        | -        | $66.2\%$ | $74.5\%$ |

---

> ### Author Response · Authors · 2023-11-20
> **Rebuttal-Part2**
>
> |                     | | Types             | None                 | FT                     | ANP                   | NAD                    | SS                    | ABL                   | EigenGuard            |
> |---------------------|-------------------|-------------------|----------------------|------------------------|-----------------------|------------------------|-----------------------|-----------------------|-----------------------|
> | GTSRB | ASR | BadNets              | $100\%$                | $0.5\%$               | $0.0\%$                | -        | -        | $1.0\%$               | $2.6\%$ |
> |                     |                   | Blend                | $100\%$                | $91\%$                | $20.7\%$               | -        | -        | $23.3\%$            | $13.7\%$ |
> |                     |                   | SIG                  | $93.8\%$               | $100\%$               | $100\%$                | -        | -        | $6.2\%$               | $4.6\%$ |
> |                     |                   | ***CLB***     | $98.8\%$  | $65.7\%$ | $16.7\%$  | -        | -        | $7.3\%$  | $5.8\%$ |
> |                     | ACC | BadNets              | $96.1\%$               | $96.7\%$              | $95.3\%$               | -        | -        | $94.7\%$              | $95.5\%$ |
> |                     |                   | Blend                | $93.4\%$               | $96.8\%$              | $93.1\%$               | -        | -        | $93.1\%$              | $94.7\%$ |
> |                     |                   | SIG                  | $95.2\%$               | $95.1\%$              | $95.2\%$               | -        | -        | $94.8\%$              | $95.5\%$ |
> |                     |                   | ***CLB***     | $95.3\%$  | $95.4\%$ | $94.3\%$  | -        | -        | $94.3\%$ | $95.1\%$ |
>
> [1] Neural Attention Distillation: Erasing Backdoor Triggers From Deep Neural Networks.
>
> [2] Spectral signatures in backdoor attacks.
>
> [3] WaNet -- Imperceptible Warping-Based Backdoor Attack
>
> [4] Invisible Backdoor Attack with Sample-Specific Triggers
>
>
>
> Apart from that, we also noticed that the spectral signature cannot perform well under the backdoor bench’s setting. The phenomenon may caused by its dataset split being parametric sensitive and the undetected few backdoor samples that are not detected can also lead to backdoor behavior in backdoor training, which also demonstrates that the detection and retraining defense methods are unstable.
>
> **3. About the novelty of our methods.**
>
> Our work is different from [1],[2]:
>
> (1) First, the spectral tools are a widely used method in analysis. Therefore, the novelty of our work and former works like you proposed are the findings, not the analysis methods. And the findings and perspectives of former works and ours are different.
>
> [1] analyzes the feature activation difference on natural and backdoor samples for a pre-trained model and [2] analyzes the gradient defense. However, our work mainly focuses on the relationship between the rank and spectrum of the trigger feature space with the attack success rate, shown in Table 2 and Figure 2. Then we conclude that trigger features’ rank is much smaller than natural features and the reason for backdoor behavior is attributed to triggers dominating the other features, which means their corresponding singular values are larger than others. Then we propose our defense method by averaging all useful features to lead the model to make the correct prediction.
>
> (2) Secondly, the goals of these works are different. [1] tries to split datasets and retrain the model. Then they need a complicated multi-stage training procedure. [2] tries to defend against the backdoor attack with additional clean data.
>
> However, our goal is more challenging. We would like to allow users to do end-to-end training without the concern of possible triggers in their datasets by just adding our module to their model.
>
> [1] Spectral signatures in backdoor attacks. Advances in neural information processing systems, 31.
>
> [2] Efficient Backdoor Removal Through Natural Gradient Fine-tuning.
>
> **4. About the threat model that our paper tries to address?**
>
> Our paper tries to achieve safe natural training against unauthorized backdoor datasets by involving a new module in the training models. With such a module, users can naturally train their models safely on unauthorized datasets without the need to adopt different methods for retraining, and purifying datasets which are both time-consuming and unstable.
>
> The majority of model users who do not know backdoor attacks will just collect data and train them without purifying the data. Our methods are trying to protect this part of users by involving our module in released ResNet codes or other model codes.

---

> ### Author Response · Authors · 2023-11-20
> **Rebuttal Part3**
>
> **5. About A and k and $\sigma_k$.**
>
> A is just a matrix for calculating effective ranks. In our following evaluations, we change a feature tensor of (batch_size, channel, height, width) to be a (batch_size, channel*height*width) matrix and calculate the effective ranks to explore the rank of the trigger feature space.
>
> k here is the index, and $\sigma_k$ means the $k$-th singular value.
>
> **6. About the CLB attack.**
>
> We also evaluate the CLB attacks on CIFAR-100 and GTSRB datasets as shown in the above table. And the results show that our methods are also effective.
>
> **7. About the improvement of the natural accuracy.**
>
> I think the improvement of the natural accuracy is caused by the EigenGuard module can also regularize the neural network by damping the top singular values. Therefore, our EigenGuard can make neural networks’ activation to be well-conditioned. It can avoid overfitting in some circumstances and lead to better results.

---

> > ### Comment · Reviewer_6rWf · 2023-11-22
> > **Thanks for the response**
> >
> > Dear authors,
> >
> > Thanks for the detailed response. I appreciate it. However, some of my concerns are still not addressed.
> >
> > First, if my understanding is correct, when comparing with new baselines, EigenGuard has a high ASR (all cases for Wanet, SSBA, and some cases for CLB. The differences are large). I am concerned if it means the proposed method cannot defense the attacks effectively, especially for stealthy attacks. If this method does not yield better defense performance, it will be important to explore other necessities for deploying this method. Also, I know that EigenGuard may provide better BA, but it is usually not the main focus of backdoor defenses.
> >
> > Secondly, the threat model means the assumptions about the attack, e.g., whether the defender needs to control to the dataset or the model itself, whether model need to be modified, etc. Including more details for the assumptions could help.

---

> ### Author Response · Authors · 2023-11-22
> **Further clarification**
>
> Thanks for your reply. Here is our further clarification:
> 1. We don't think the success rate for new attacks and CLB attacks is high. The attack success rate shows that our model's prediction is worse than the random prediction, even the worst result SSBA is also around the random guess (10% for CIFAR-$10$). Therefore, we think the defense is successful. In the meantime, I think natural accuracy is also important because it means the utility of the models. For example, all defense methods show a clear drop in the CIFAR-100 dataset, which means these models are somehow useless.
> 2. About threat models, the defender can only modify their model architecture, they cannot modify the training code or datasets. Such a scenario is untouched but also realistic. For example, in distributed training, other participants cannot control the datasets or the training procedure. Furthermore, in most cases,  many users won't purify their downloaded datasets and do complicated multi-stage training. Then adding our modules can make them free from backdoor threats. Since our module shows no clear degeneration in models' natural performance on various models, such a method can be directly used in any model's code like BatchNorm or ReLU.

---

> > ### Comment · Reviewer_6rWf · 2023-11-23
> > **Response to the rebuttal**
> >
> > Dear Authors,
> >
> > Thank you very much for your quick response. However, after careful consideration, I think the novelty and evaluation of the paper are still not enough for me to give a clear acceptance. I think there are two main problems. One is that other work has already proposed similar concepts and the additional theoretical analysis for this paper is too abstract. The second problem is on the evaluation setting. While the rebuttal provides results on more baselines, the evaluation scale is limited and the performance against recent attacks is concerning. The evaluation results are also unstable, making me wonder the generalization of the proposed method.

---

### Official Review · Reviewer_fVxx · 2023-11-04

**Soundness:** 2 fair
**Presentation:** 2 fair
**Contribution:** 2 fair
**Rating:** 3
**Confidence:** 5

**Summary:**

This paper proposes a new defense mechanism against backdoor attacks. The paper first investigates the singular value decomposition of the of the activation layer of neural networks. These investigations reveal that the dominant singular values of the activation layer preserve relative information of the clean vs. backdoor data, while the low-energy singular values mix them together. Motivated by this observation, this paper proposes EigenGuard. In short, EigenGuard uses a spectral filter to lower the significance of the low-energy singular values to combat backdoor attacks. Experimental results over CIFAR-10, CIFAR-100, and GTSRB shows that the proposed method effectively combat backdoor attacks such as BadNets, Blend, SIG, and Clean Label attacks.

**Strengths:**

- This work presents interesting observations regarding the influence of backdoor attacks on the singular value decomposition of neural network activation layers.

- The experimental results demonstrate that this approach can be useful in combating some existing backdoor attacks.

**Weaknesses:**

- The paper starts its discussions in the introduction by stating inaccurate facts about the state of existing backdoor defenses. In particular, the paper says: "_When attempting to train a clean model on unauthorized datasets, existing methods typically try to
fine-tune the neural networks on some additional datasets..._" While this was the case for older backdoor defense methods, recently, there has been quite a good progress in proposing methods that do not necessarily require clean held-out validation set to mitigate backdoor attack. For example, see [1-4]. Additionally, saying that "_With the uncontaminated datasets split after the detection, we can train the model to unlearn backdoor triggers with designed unlearning loss._" about Spectral Signatures (Tran et al. [1]) is inaccurate. We know that this method has a two-step training process, where after filtering the poisoned data, it re-trains the entire network and, as such, has less effect on the benign accuracy. These ambiguities in the presence of the past literature has led the paper to claim in Table 1 that it is the only work that doesn't require clean data AND doesn't do unlearning AND uses the natural training. There are other works within the literature that satisfy this criteria. For example, see [2-4].

- There are several major issues with the current method:

   1. The paper emphasizes over and over about the relationship of the backdoor triggers with a low-rank space. For instance, it reads: "_One plausible explanation for this observation is the limited effective subspace associated with the trigger. This suggests that the trigger features are distributed in a low-dimensional subspace._" However, these explanations are just a restatement of the actual method used in the paper, not based on step-by-step intuitive reasonings. I highly encourage the authors to re-write these statements and try not to rely on the observations in the figures but on the intuitions and explain why this method should work.

   2. More importantly, the observations made in Figure 2, the explanation of the paper about this figure, and the actual methodology seem different. In particular, after plotting the t-SNE of different singular values in Figure 2, the paper says: "_However, from the middle t-SNE figure, one can see that the pink dots represent backdoor images distributed uniformly in the space and overlap with other color dots. Thereby, the network cannot classify these samples as trigger classes since they are similar to samples belonging to different natural classes._" So, from this explanation it seems that the natural way of dealing with backdoors is to remove the dominant singular values. However, as shown in Algorithm 1 in the Spectral Filter, the proposed method actually preserves those singular values and dampens the effects of the remaining ones. Perhaps there is a misunderstanding here that needs to be resolved.

  3. There are two important related works that this paper needs to discuss its relationship with them in more detail. First, the method of Spectral Signatures [1] also uses SVD in the feature space of neural nets to filter samples that are poisoned. Second, Collider [2] uses local intrinsic dimensionality (LID) to argue that backdoors reside in a locally high-dimension manifold. The current work argues that backdoors reside in a low-dimensional sub-space (even though it does it rather informally) and as such, it is vital to clarify its stance with prior work.

  4. The theoretical contributions seem too abstract. In other words, it is unclear how the provided theory is supporting the proposed method, as having a set of segregated feature vectors for backdoored vs. clean data seems too artificial and not directly related to EigenGuard. Please re-write this section to make its connections with the proposed method clearer.

- The experimental results are limited. The paper tests its approach against BadNets, Blend, SIG, and Clean Label attacks that are published before 2020s and declares that they are state-of-the-art. There are newer backdoor attacks that exist in the literature today, including but not limited to: Dynamic Attacks [6], WaNet [7], ISSBA [8], Refool [9], etc. The same also applies to the baseline defenses used in the paper, where many recent works, such as [1-4], are left out. To provide a comprehensive evaluation, incorporating all these newer baselines is necessary. Preferably, large-scale experiments on ImageNet dataset is also needed.

[1] Tran et al. "Spectral signatures in backdoor attacks." _NeurIPS_, 2018.

[2] Dolatabadi et al. "Collider: A robust training framework for backdoor data." _ACCV_, 2022.

[3] Hayase et al. "Spectre: Defending against backdoor attacks using robust statistics." _ICML_, 2021.

[4] Liu et al. "Beating Backdoor Attack at Its Own Game." _ICCV_, 2023.

[5] Huang et al. "Distilling Cognitive Backdoor Patterns within an Image." _ICLR_, 2023.

[6] Nguyen et al. "Input-aware dynamic backdoor attack." _NeurIPS_, 2020.

[7] Nguyen et al. "Wanet–imperceptible warping-based backdoor attack." _ICLR_, 2021.

[8] Li et al. "Invisible backdoor attack with sample-specific triggers." _ICCV_, 2021.

[9] Liu et al. "Reflection backdoor: A natural backdoor attack on deep neural networks." _ECCV_, 2020.

**Questions:**

Apart from the questions raised above, here are some additional questions/suggestions:

- What settings are used for the empirical evaluations of Section 3? Some figures, such as Figure 2, only present the result without mentioning the dataset, backdoor, model architecture, etc.

- Does the same empirical analysis (those in Section 3) also hold for ALL the above-mentioned attacks [6-9]? Does it also hold for clean-label attacks?

- What is the significance of the theoretical analysis, and how does it relate to the rest of the paper? Can you verify its statements in a realistic setting with quantitative analysis?

- Using the term "head" for the first layers of the ResNet model is confusing. Usually, head refers to the last classification layer and is a short-term for "classification head". Consider using feature extractor or other alternatives to avoid confusion.

---

> ### Author Response · Authors · 2023-11-20
> **Rebuttal-Part1**
>
> Thanks for your review, the following is our response.
>
> **1. About the claims in Table 1**
>
> We need to claim that our method tries to train the neural networks in an end-to-end procedure. It is a challenging setting. The goal of our paper is to allow users to do end-to-end training without the concern of possible triggers in their datasets by just adding our module to their model.
>
> Although some methods like [1] which split the datasets and then naturally retrain the neural networks will also not influence the natural accuracy much, it is not a natural training way for neural networks. Such methods usually need a few different training stages, which is still a little complicated. Thereby, many users may still use the end-to-end training procedure instead of methods like [1]. To make it more clear, we also rewrite the tables.
>
> **2. About “However, these explanations are just a restatement of the actual method used in the paper, not based on step-by-step intuitive reasonings. I highly encourage the authors to re-write these statements and try not to rely on the observations in the figures but on the intuitions and explain why this method should work.”**
>
> Our method is proposed based on our findings on the features’ eigenspace during training on different samples and tries to propose a new module to avoid such behaviors.
>
> First, we find that the trigger features can only concentrated in a low-rank subspace according to Figure 1 and Table 2. Then we find that trigger features are mainly distributed in the top singular space in Figure 2, which means the backdoor predictions are made because trigger features contribute a lot to the prediction due to their large singular value.
>
> Due to these findings, we propose our EigenGuard which damps the magnitude of trigger features while involving more natural features for prediction. Then we can defend against several backdoor attacks without any additional treatments during training. It’s a natural finding to solution way for research.
>
> **3.About the “inconsistency“ between Figure 2 and our method:**
>
> Here we only damp top eigenvalues instead of dropping them because $\sigma_k$ is much smaller than $\sigma_1$, which is consistent with our findings in Figure 2. From Figure 2(a), one can see that the trigger features are large enough to dominate the classification of backdoor samples, so making it smaller can decrease its impact. In other words, we try to average the backdoor features in Figure 2 (a) and (b) to make the backdoor samples hard to classify.
>
> The reason why we not simply removing the top eigenvalues is such a method will make natural classes hard to classify as we can see in Figure 2(b) and lead to a huge drop in natural accuracy as listed in the following experiments on CIFAR-10 with BadNets:
>
> | Methods | ACC. | ASR. |
> | -------- | ------- | ------- |
> | Remove top 10 Singulars  |  $85.4\%$   |  $4.2\%$   |
> | Ours | $92.6\%$     |  $5.3\%$   |
>
> Also, as shown in Figure 4, the EigenGuard module can make the features perform as we wish.
>
> **4. About our methods’ relationship with Spectral Signature and Local intrinsic dimensionality.**
>
> There are two differences between ours and theirs:
>
> (1) Our method is a plug-in module for neural architectures, neural networks can naturally defend against some backdoor attacks with end-to-end natural training by inserting our EigenGuard model into the model’s architecture. The other two methods only try to use the poisoned model's behavior to detect the trigger images and then try to purify the datasets to prevent backdoor attacks, which need complicated post-stage retraining stages.
>
> (2) Although both works analyze the feature behaviors, ours tries to detect the feature’s spectral influence during training while others only analyze the model's feature behavior after training.
>
> **5. About the theory:**
> Our theory tries to demonstrate that damping the dominating trigger features will make the models use natural features to make correct predictions. It is also consistent with our method, we make the top singular values smaller to avoid the triggers dominating classifications.

---

> ### Author Response · Authors · 2023-11-20
> **Rebuttal -- Part 2**
>
> **6. About more evaluations:**
>
> We also add some evaluations for our methods, we list some results below and you may find more results on the revision with SSBA[2], WaNet[3] attack and NAD[4], Spectral Signature defense using backdoor bench[6]’s implementation on CIFAR-$10$ dataset due to the time limits. The results show that our method can perform well on different attacks.
>
> Apart from that, we also noticed that the spectral signature cannot perform well under the backdoor bench’s setting. The phenomenon may caused by its dataset split being parametric sensitive and the undetected few backdoor samples that are not detected can also lead to backdoor behavior in backdoor training, which also demonstrates that the detection and retraining defense methods are unstable.
>
> |                     | | Types             | None                 | FT                     | ANP                   | ***NAD***                    | ***SS***                    | ABL                   | EigenGuard            |
> |---------------------|-------------------|-------------------|----------------------|------------------------|-----------------------|------------------------|-----------------------|-----------------------|-----------------------|
> | C10                 | ASR               | BadNets              | $100\%$                | $3.0\%$               | $0.5\%$                | $6.7\%$  | $99.7\%$ | $3.1\%$  | $5.3\%$             |
> |                     |                   | Blend                | $100\%$                | $10.2\%$              | $0.5\%$                | $3.8\%$  | $100\%$  | $15.2\%$ | $0.4\%$             |
> |                     |                   | CLB                  | $100\%$                | $1.2\%$               | $4.0\%$                | $21.7\%$ | $88.3\%$ | $0.1\%$ | $1.0\%$              |
> |                     |                   | SIG                  | $94.2\%$               | $0.4\%$               | $0.3\%$                | $6.8\%$  | $89.1\%$ | $0.01\%$   | $2.8\%$           |
> |                     |                   | ***WaNet***                | $92.3\%$  | $15.1\%$ | $1.7\%$   | $28.9\%$ | $90.1\%$ | $2.3\%$  | $8.8\%$|
> |                     |                   | ***SSBA***                | $100\%$   | $23.7\%$ | $0.9\%$   | $100\%$  | $88.3\%$ | $4.4\%$  | $17.1\%$|
> |                     | ACC               | BadNets              | $93.7\%$               | $87.2\%$              | $90.2\%$        x       | $90.1\%$ | $92.3\%$ | $89.1\%$  | $92.6\%$             |
> |                     |                   | Blend                | $94.8\%$               | $88.9\%$              | $93.4\%$               | $93.3\%$ | $93.1\%$ | $88.7\%$  | $93.5\%$            |
> |                     |                   | CLB                  | $93.8\%$               | $91.9\%$              | $92.7\%$               | $91.8\%$ | $92.4\%$ | $89.3\%$  | $93.3\%$            |
> |                     |                   | SIG                  | $93.6\%$               | $91.6\%$              | $93.4\%$               | $92.1\%$ | $91.9\%$ | $89.0\%$  | $92.9\%$            |
> |                     |                   | ***WaNet***                | $92.7\%$  | $93.5\%$ | $90.5\%$  | $92.2\%$ | $87.4\%$ | $88.5\%$ | $92.7\%$|
> |                     |                   | ***SSBA***    | $92.8\%$  | $89.7\%$ | $88.7\%$  | $92.3\%$ | $71.5\%$ | $85.6\%$ | $93.1\%$|
>
> [1] Spectral signatures in backdoor attacks. NeurIPS, 2018.
>
> [2] Invisible Backdoor Attack with Sample-Specific Triggers ICCV 2021
>
> [3] WaNet -- Imperceptible Warping-Based Backdoor Attack ICLR 2021
>
> [4] Neural Attention Distillation: Erasing Backdoor Triggers From Deep Neural NetworksICLR 2021

---

### Meta-Review · Area_Chair_XvxR · 2023-12-05

**Metareview:**

This paper introduces EigenGuard, a novel defense method against backdoor attacks, drawing on insights from the spectral characteristics of features. However, the applicability of the proposed defense method to a wider range of backdoor attacks remains unclear. The reviewers raised several issues: (i) missing discussion of relevant existing literature; (ii) theoretical contributions that don't fully substantiate the paper's claims; and (iii) experiments that do not account for the latest state-of-the-art attacks. Although the authors participated in a constructive discussion during the rebuttal phase, not all concerns were satisfactorily addressed. The overall agreement among the reviewers is that this paper has shortcomings in soundness and clarity. Moreover the impact of the contributions is limited. Thus, I advise against accepting this paper.

**Justification For Why Not Higher Score:**

The agreement among the reviewers is clear.

**Justification For Why Not Lower Score:**

N/A

---

### Decision · Program_Chairs · 2024-01-16

Reject